# A Meta-Analysis to Assess the Efficacy of HER2-Targeted Treatment Regimens in HER2-Positive Metastatic Colorectal Cancer (mCRC)

**Akshit Chitkara** [1], **Muhammad Bakhtiar** [2], **Ibrahim Halil Sahin** [3], **Dennis Hsu** [3], **Janie Zhang** [3], **FNU Anamika** [4], **Mahnoor Mahnoor** [5], **Rabeea Ahmed** [2], **Sepideh Gholami** [6] and **Anwaar Saeed** [3,*]

[1]   Department of Internal Medicine, University of California, Riverside, CA 92521, USA; akshit.chitkara@medsch.ucr.edu

[2]   School of Medicine, King Edward Medical University/Mayo Hospital, Lahore 54000, Pakistan; hussainbakhtiar77@gmail.com (M.B.); ahmedra99@gmail.com (R.A.)

[3]   Department of Medicine, Division of Hematology-Oncology, University of Pittsburgh Medical Center (UPMC), Pittsburgh, PA 15232, USA; sahinih@upmc.edu (I.H.S.); hsudj@upmc.edu (D.H.); zhangjy2@upmc.edu (J.Z.)

[4]   Department of Internal Medicine, Hackensack Meridian Ocean University Medical Center, Brick, NJ 08724, USA; fnu.anamika@hmhn.org

[5]   School of Medicine, Mohtarma Benazir Bhutto Shaheed Medical College, Mirpur 10230, Pakistan; mahnoor2653@gmail.com

[6]   Department of Hematology-Oncology, Northwell Health Cancer Institute, New Hyde Park, NY 11042, USA; sgholami@northwell.edu

*   Correspondence: saeeda3@upmc.edu; Tel.: +1-412-623-2091

**Abstract:** Recent trials provide evidence that HER2 is a potential new target for patients with colorectal cancer. While HER2-positive tumors do not show a very encouraging response to anti-HER2-positive agents like trastuzumab alone, promising results have been observed when combined with other synergistically acting tyrosine kinase inhibitors (TKIs). Our meta-analysis was conducted following the Cochrane Handbook and written following the PRISMA guidelines. The protocol was registered on PROSPERO with the registration number CRD42022338935. After a comprehensive search for relevant articles, 14 CTs were identified and uploaded to Rayyan, and six trials were ultimately selected for inclusion. The meta-analysis revealed that a median of three prior lines of therapy was used before enrolling in the six trials comprising 238 patients with HER2-positive metastatic colorectal cancer (mCRC). The pooled objective response rate (ORR) and disease control rate (DCR) were 31.33% (95% confidence interval [CI] 24.27–38.39) and 74.37% (95% CI 64.57–84.17), respectively. The pooled weighted progression-free survival (PFS) was 6.2 months. The pooled ORR and DCR meta-analysis indicate a significant response to HER2-targeted therapy in this patient in HER2-positive mCRC. Additionally, a pooled PFS of 6.2 months suggests that HER2-targeted treatment regimens are associated with a meaningful improvement in survival outcomes in this population.

**Keywords:** trastuzumab deruxtecan; HER2-positive; tyrosine kinase inhibitors; metastatic colorectal cancer

## 1. Introduction

Colorectal cancer (CRC) is the second leading cause of cancer-related deaths globally, according to the World Health Organization (WHO), and the third most common cancer diagnosed in the United States [1]. Although effective cancer screening measures have decreased CRC incidence and mortality rates, there has been a recent rise in the number of young patients diagnosed with colon cancer [2–6]. Only 20–30% of CRC is associated with hereditary syndromes caused by highly penetrant autosomal dominant and recessive mutations [7].

The standard treatment for CRC, like most tumors, involves surgery, chemotherapy, and radiotherapy [8–12]. Emerging treatment options such as laparoscopic resection, neoadjuvant treatment followed by surgery, and systemic chemotherapy provide additional avenues for patients to pursue a cure. However, these novel therapies have a limited impact on cure rates and long-term survival [13]. Various treatment modalities are under investigation, including checkpoint inhibitors, cancer vaccines, adoptive cell transfer, oncolytic virus therapy, and other agents, focusing on immune checkpoint inhibitors [14]. Recent trials provide evidence for HER2 as a potential new target for patients with colorectal cancer [15–18].

A small subset of patients has a HER2-positive oncogene expressing CRC, allowing targeted therapy [19,20]. In total, 32% of HER2-positive CRCs have short variant alterations not detectable via routine immunohistochemistry or fluorescence in situ hybridization testing. Ongoing clinical trials indicate promising results for anti-HER2 therapies [21]. While HER2-positive tumors do not show a very encouraging response to anti-HER2-positive agents like trastuzumab alone, promising results have been observed when combined with other synergistically acting TKIs [22–24].

Our meta-analysis aims to comprehensively compile information on newly studied targeted therapies, including trastuzumab in combination with TKIs, or trastuzumab-based antibody–drug conjugate (ADC) regimens, with an emphasis on their merits, demerits, and most common adverse effects. This article underscores the need for further investigation into different HER2-targeting treatment modalities for HER2-positive CRC.

## 2. Materials and Methods

Our meta-analysis was conducted in accordance with the Cochrane Handbook for systematic reviews of interventions [25]. It was written following the PRISMA guidelines [26]. The protocol was registered on PROSPERO with the registration number CRD42022338935.

### 2.1. Eligibility Criteria

This study included trials involving patients with (1) HER2-positive mCRC, (2) of any age, (3) any sex, and (4) from any geographical area. (5) We focused our study on clinical trials. (6) The therapeutic agent used in these trials must be an anti-HER2 agent, such as trastuzumab, with a TKI or ADC. We excluded studies that involved (1) non-HER2-positive CRC, (2) all other solid tumors, (3) systematic reviews, meta-analyses, or papers other than clinical trials (CTs), and (4) trials that did not have a drug targeting an anti-HER2 agent.

### 2.2. Information Sources

Studies were selected by searching through electronic databases and clinical trial registries. Electronic databases included CENTRAL, MEDLINE (via PubMed), and Embase. Clinical trial registries included Clinicaltrials.gov and the WHO international registry of trials. We started to search for relevant trials on 25 June 2022.

### 2.3. Search Strategy

The terms used to search through the databases included "anti-HER2-positive agents and HER2-positive CRC", "trastuzumab plus adjuvants against CRC," "treatment of HER2-positive CRC", "HER2-positive CRC management", "trastuzumab deruxtecan against HER2-positive CRC", "trastuzumab and tyrosine kinase inhibitors," "antibody-drug conjugates against HER2-positive CRC" and "antibody-drug conjugates and tyrosine kinase inhibitors".

### 2.4. Study Selection

After conducting a comprehensive search for relevant articles, 14 CTs were identified and uploaded to Rayyan. Following de-duplication using the inbuilt software of Rayyan, 12 articles were incorporated into the final analysis.

Multiple reviewers conducted a simultaneous review of the articles, and two articles were found to be duplicates. One article was deemed irrelevant and excluded. Of the remaining nine, three contained only preliminary data. We could retrieve data from one of these studies by contacting the researchers, but the researchers of the remaining two studies did not respond, so their data were discarded. One trial was a short-term outcome of another long-term study and was removed as a duplicate.

After reading the full-length papers, six trials were ultimately selected for inclusion in our analysis. Any discrepancies during the screening process were resolved through mutual discussion among the reviewers. The details of the screening process are provided in the PRISMA flow sheet, as shown in Figure 1.

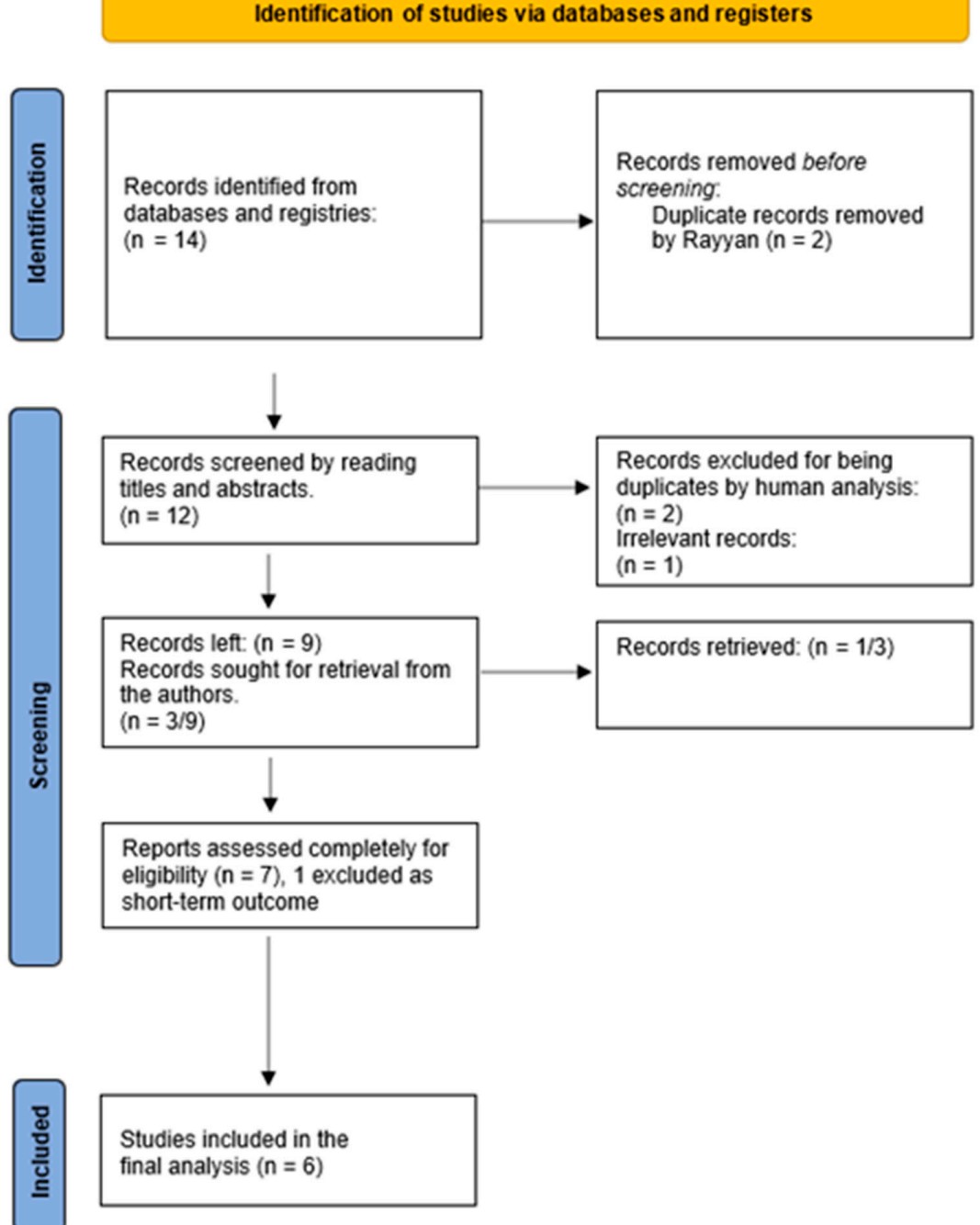

**Figure 1.** Study selection algorithm based on PRISMA 2020 guidelines. (n = Number of studies).

### 2.5. Data Collection Process

After a review of the trials, data were extracted and saved on an Excel sheet. There were two types of data: (1) information like author ID, study design, the type of intervention, and the number of patients; (2) outcome variables like the objective response rate (ORR), disease control rate (DCR), and other relevant primary and secondary outcomes.

### 2.6. Primary Outcome

The primary outcome of this study was based on the efficacy of the study drug judged by the following variables: ORR, DCR, and PFS. We also enlisted the complete response to the drug (CR), partial response to the drug (PR), and stable disease (SD); the disease has neither progressed nor regressed and progressive disease (PD).

### 2.7. Secondary Outcome

Any additional effects of the study drug on patients that were not part of the primary outcome but deemed appropriate as an outcome, such as adverse events, were considered secondary outcomes of the study.

### 2.8. Quality Assessment

The RoB (Risk of Bias) tool developed by the Cochrane Library of Systematic Reviews was used for quality assessment [27,28]. NHLBI (NIH) tools assessed trials involving no control group for quality assessment. Each trial was judged by two authors independently, and any conflicts were resolved through mutual discussion (Table 1).

**Table 1.** Assessment of trials using the NHLBH tool.

| Criteria | Tsurutani et al. [29] | Bianchi et al. [30] | Tosi et al. [31] | Sienna et al. [32] |
|---|---|---|---|---|
| Question objective clearly stated? | Yes | Yes | Yes | Yes |
| Are eligibility criteria prespecified? | Yes | Yes | Yes | Yes |
| Population in the study representative of the target population? | Yes | Yes | Yes | Yes |
| Were all eligible participants enrolled? | Yes | Yes | Yes | Yes |
| Was the sample size sufficiently large? | Yes | Yes | Yes | Yes |
| Was the intervention clearly described and delivered consistently throughout the trial? | Yes | Yes | Yes | Yes |
| Outcome measures prespecified, clearly defined, reliable, and assessed consistently? | Yes | Yes | Yes | Yes |
| Were the people assessing outcomes blinded? | Yes | Yes | N/A | Yes |
| Was lost to follow up 20% or less? | Yes | Yes | Yes | Yes |
| Were those lost to follow-up accounted for in the analysis? | Yes | Yes | Yes | Yes |
| Were statistical methods done that gave $p$-value? | N/A | Yes | N/A | N/A |
| Was an interrupted time series design used? | Yes | Yes | N/A | N/A |
| Was the study at a group level (e.g., the whole hospital)? | No | No | No | No |
| If yes, did the study analysis consider individual-level data to determine effects at the group level? | No | No | No | No |
| Quality | High | High | Some concern | Some concern |

The quality assessment results show that most of the studies are of high quality. Only two studies have some level of concern. N/A = Not applicable.

*2.9. Data Synthesis*

Quantitative data were extracted and collected in a tabulated manner in an Excel sheet, and based on the tabulated data, a meta-analysis was performed. A random-effects model was used for DCR and ORR, and statistical heterogeneity was indicated by a *p*-value < 0.05.

In single-arm studies, there is only one intervention group, so comparing the risk of an event between two groups is impossible. Therefore, hazard ratio (HR) is not applicable in single-arm studies. Still, other statistical measures, such as the ORR, DCR, and PFS, are commonly used to evaluate the effectiveness of treatment. These measures can provide valuable information about the efficacy of treatment, but they do not allow for a direct comparison between different groups. It is important to note that single-arm studies have some limitations, and their results must be interpreted cautiously. Without a comparison group, it is difficult to determine whether any observed treatment effects are due to the treatment itself or other factors such as natural disease progression, regression to the mean, or placebo effects. Therefore, single-arm studies are often followed by more extensive randomized controlled trials to confirm the efficacy and safety of treatment [33].

We calculated the pooled ORR for these six single-arm studies through the following steps:

1. Calculate the weighted average of ORR: calculate the weighted average of ORR by weighing the ORR estimates from each study by their sample size using the formula:

$$\text{weighted average ORR} = \Sigma \, (\text{ORR\_i} \times \text{weight\_i}) \tag{1}$$

2. Calculate the standard error of the weighted average of ORR: calculate the standard error of the weighted average of ORR using the formula:

$$SE = \sqrt{(\Sigma(w_i \times (1 - ORR_i) \times ORR_i)/\Sigma(w_i \times n_i))} \tag{2}$$

where $w_i$ is the weight assigned to each study (i.e., the study's sample size divided by the total sample size), $ORR_i$ is the ORR estimate for each study, and $n_i$ is each study's sample size.

3. Calculate the 95% confidence interval: calculate the 95% confidence interval for the weighted average of ORR using the formula:

$$CI = ORR \pm (1.96 \times SE) \tag{3}$$

ORR is the weighted average of ORR, and SE is the standard error of the weighted average of ORR.

4. The pooled ORR provides an overall estimate of the treatment effect in single-arm studies and can be used to inform clinical decision making and guide further research. Nonetheless, it is essential to emphasize that the pooled ORR is only as valid as each study's individual ORR estimates and may be subject to confounders or biases.

This methodology was also utilized to estimate our study's DCR and PFS. Two other reviewers validated all the data extraction and calculations. Additionally, interpreting these measures may require clinical expertise and careful consideration of the study design and patient population.

## 3. Results

*3.1. Study Characteristics*

Two of the six trials selected evaluated trastuzumab deruxtecan, and one evaluated trastuzumab plus lapatinib. One trial evaluated pertuzumab plus trastuzumab emtansine, while one evaluated trastuzumab plus tucatinib, and one evaluated trastuzumab plus pyrotinib (Table 2). Regarding prior lines of treatment, the median number of prior lines was two and three in two trials and four and five in one trial each. The pooled median number of prior lines in therapy was estimated to be three before enrollment in the trial. Adverse

events and safety were also assessed as secondary outcomes of the study. The results have been divided into subheadings. This portray a concise analysis of the experimental results and their interpretation.

**Table 2.** Characteristics of the studies selected.

| Author ID | Study Design | Intervention | Year of Publication | No. of Subjects | Prior Line of Rx | HER2 Mutation | RAS Mutation | BRAF Mutation |
|---|---|---|---|---|---|---|---|---|
| Tsurutani et al. [29] | Non-randomized phase 1 dose expansion clinical trial | Trastuzumab deruxtecan | March 2020 | 20 | 4 | 5 Kinase domain, 1 Transmembrane domain, and 0 Extracellular domain | 5 KRAS and 2 NRAS | - |
| Fu et al. [34] | Non-randomized phase 2 trial | Trastuzumab + Pyrotinib | March 2023 | 18 | 2 | 5 HER2 | 12 RAS wild-type, 5 KRAS and 1 NRAS | - |
| Bianchi et al. [30] | Single arm, multicenter, phase 2 clinical trial | Pertuzumab + Trastuzumab emtansine | January 2020 | 31 | 3 | - | - | - |
| Strickler et al. [35] | Open-label phase 2 clinical trial | Trastuzumab + Tucatinib | January 2023 | 84 | 3 | - | - | - |
| Tosi et al. [31] | Open-label Phase 2 Non-randomized | Trastuzumab + Lapatinib | January 2020 | 32 | 5 | - | 32 KRAS exon 2 (codons 12 and 13) wild-type | - |
| Siena et al. [32] | Open-label Phase 2 Non-randomized | Trastuzuma deruxtecan | June 2023 | 53 | 2 | - | 52 RAS wild-type and 1 NRAS | 53 BRAF wild-type |

## 3.2. Result of Synthesis

A narrative synthesis was conducted using the data collected from the selected studies. The CR, PR, SD, and PD of the selected studies are shown in Table 3, highlighting the individual study results with the maximum number of patients achieving stable disease in the Bianchi et al. study group [30]. The results of the reviewed trials demonstrate that trastuzumab plus tucatinib and trastuzumab deruxtecan exhibit a promising ORR of >30%, with the latter showing slightly better results than the former [35]. Combining lapatinib and pyrotinib with trastuzumab resulted in an ORR of 28.12% and 22.2%, respectively [31,33]. Trastuzumab emtansine plus pertuzumab had a lower ORR of only 9.68% [30]. See Table 4 for further details. All ADC and TKI-involved regimens listed in Table 4 achieved an effective DCR. Trastuzumab deruxtecan had the most effective DCR (>80%), followed by trastuzumab emtansine plus pertuzumab (77.42%) and trastuzumab plus tucatinib (71.43%) [29,30,35].

**Table 3.** The CR, PR, SD, and PD of the studies selected.

| Author ID/IDs | Drug Combination | CR | PR | SD | PD |
|---|---|---|---|---|---|
| Tsurutani et al. [29] (20 patients) | Trastuzumab deruxtecan | 0% (0/20) | 15% (3/20) | 65% (13/20) | 15% (3/20) |
| Sienna et al. [32] (53 patients) | Trastuzumab deruxtecan | 2% (1/53) | 43.40% (23/53) | 37.73% (20/53) | 9% (5/53) |
| Fu et al. [34] (18 patients) | Trastuzumab + Pyrotinib | 0% (0/18) | 22.22% (4/18) | 38.89% (7/18) | No data available |
| Bianchi et al. [30] (31 patients) | Trastuzumab emtansine + Pertuzumab | 0% (0/31) | 9.68% (3/31) | 67.74% (21/31) | 22.58% (7/31) |
| Strickler et al. [35] (84 patients) | Trastuzumab + Tucatinib | 3.57% (3/84) | 34.52% (29/84) | 33.33% (28/84) | 26.19% (22/84) |
| Tosi et al. [31] (32 patients) | Trastuzumab + Lapatinib | 3.12% (1/32) | 25% (8/32) | 40.62% (13/32) | No data available |

(CR = Complete response, PR = Partial response, SD = Stable disease, PD = Progressive disease).

**Table 4.** ORR, DCR, and PFS of the studies selected.

| Author | Drug Combination | ORR (95%CI) | DCR (95%CI) | PFS (Months) |
|---|---|---|---|---|
| Tsurutani et al. [29] (20 patients) | Trastuzumab deruxtecan | 15% CI 3.2–37.9 (3/20) | 80% CI 56.3–94.3 (16/20) | 4.1 (2.1–5.9) |
| Sienna et al. [32] (53 patients) | Trastuzumab deruxtecan | 45.28% CI 31.6–59.6 (24/53) | 83.01% CI 70.2–91.9 (44/53) | 6.9 (4.1 to NE) |
| Fu et al. [34] (18 patients) | Trastuzumab + Pyrotinib | 22.2% CI 6.4–47.69 (4/18) | 61.11% CI 35.8–82.7 (11/18) | 3.4 (1.8–4.3) |
| Bianchi et al. [30] (31 patients) | Trastuzumab emtansine + Pertuzumab | 9.68% (3/31) | 77.42% (24/31) | 4.1 (3.6–5.9) |
| Strickler et al. [35] (84 patients) | Trastuzumab + Tucatinib | 38.10% (32/84) | 71.43% (60/84) | 8.2 |
| Tosi et al. [31] (32 patients) | Trastuzumab + Lapatinib | 28.12% (9/32) | 68.75% (22/32) | 4.7 (3.7–6.1) |
| Cumulative weighted Meta-analysis | Pooled: a. ORR with 95% CI b. DCR with 95% CI c. PFS | a. 31.33% (95% CI 24.27–38.39) | b.74.37% (95% CI 64.57–84.17) | c. 6.2 months |

(ORR = Overall response rate, DCR = Disease control rate, PFS = Progression-free survival).

In Bianchi et al., patients with tumors displaying a higher HER2 IHC score (3+) had better PFS compared to those with a lower score (2+); patients with a score of 3+ had a PFS of 5.7 months, while those with a score of 2+ had a PFS of 1.9 months [30]. A higher HER2 IHC score was associated with a better objective response and long-lasting disease stabilization [30].

In the study Siena et al. recently published in June 2023 showed that more patients with high HER2 expression levels (IHC3+) had an objective response than those with IHC2+ and ISH-positive tumors [32]. However, the authors concluded that further studies were needed due to the low number of patients enrolled. Additionally, the authors stated that although trastuzumab deruxtecan showed antitumor activity in HER2-low breast tumors, it did not respond in patients with HER2-low metastatic colorectal cancer tumors. The study included one patient with an NRAS mutation that showed minimal changes in tumor size from the baseline. Another study by Tsurutani et al. included five patients with KRAS mutations and two patients with NRAS mutations but did not report separate outcomes for this subgroup of patients [29].

Fu et al. [34] evaluated eighteen patients for efficacy in wild-type RAS/BRAF patients. The ORR was found to be 33.3% (95% CI 13.8–60.9), and the DCR was found to be 83.3% (95% CI 51.6–97.9). A phase II basket study of trastuzumab plus pertuzumab, named MyPathway, suggested that patients with KRAS gene mutation decreased PFS (KRAS mutated: KRAS wild, found to be 1.40 months:5.30 months) and the OS (KRAS mutated: KRAS wild, found to be 8.50 months:14.00 months) compared to those with KRAS-wildtype tumors. Among patients with RAS wild type, 33.3% achieved an objective response in line with previous studies of other dual-HER2 therapies in which RAS wild-type, HER2-positive mCRC patients achieved an ORR of 30–40%. In comparison, none of the six patients expressing the RAS mutation showed an objective response in this study. Solely one patient showed stable disease, indicating that RAS predicted no clinical response to dual HER2-targeted therapy. In Tosi et al., all 32 patients had a histologically confirmed

diagnosis of mCRC with KRAS exon 2 (codons 12 and 13) wild-type status and HER2 positivity [31]. The study results detailed in the table apply to all KRAS wild-type patients.

The meta-analysis revealed that a median of three prior lines of therapy was used before enrolling in the trial. Outcome data were available for all six studies, comprising 238 patients with HER2-positive metastatic colorectal cancer (mCRC) who received HER2-targeted treatment regimens. The pooled ORR and DCR were 31.33% (95% confidence interval [CI] 24.27–38.39) and 74.37% (95% CI 64.57–84.17), respectively. The pooled weighted PFS was 6.2 months. These findings suggest that HER2-targeted treatment regimens improve PFS and lead to a higher ORR and DCR than chemotherapy in patients with HER2-positive mCRC.

### 3.3. Adverse Events

Our meta-analysis investigated the adverse effects of all drug combinations. Among patients receiving trastuzumab deruxtecan, the most common adverse events were nausea and vomiting (64.3%) [29]. Fatigue was the most common adverse event in the trastuzumab emtansine plus pertuzumab group (18%) [30]. By contrast, diarrhea was the most common adverse event in the trastuzumab plus tucatinib (52.3%) and trastuzumab plus lapatinib (84.37%) groups [31,35]. The analysis confirmed that no significant serious adverse events were noted among these patients. Other side effects, such as thrombocytopenia, were observed in 8% to 15% of patients and pruritus in 8% to 10% across various groups. Fatigue, nausea, vomiting, and dermatitis were observed in all drug combinations, and diarrhea, hyperbilirubinemia, thrombocytopenia, and pruritus were also reported (see Table 5).

**Table 5.** Adverse events of the drug combinations selected.

| Adverse Events | Drug Combination | | | | |
| --- | --- | --- | --- | --- | --- |
| | Trastuzumab Deruxtecan | Trastuzumab Emtansine + Pertuzumab | Trastuzumab + Tucatinib | Trastuzumab + Lapatinib | Trastuzumab + Pyrotinib |
| Fatigue | 34.25% (25/73) | 19.35% (6/31) | 28.57% (24/84) | 59.37% (19/32) | 38.88% (7/18) |
| Nausea and Vomiting | 64.38% (47/73) | 9.68% (3/31) | 19.04% (16/84) | 46.87% (15/32) | 38.88% (7/18) |
| Diarrhea | 28.3% (15/53) * | - | 52.38% (44/84) | 84.37% (27/32) | 94.44% (17/18) |
| Dermatitis | 5% (1/20) ** | 6.45% (2/31) | 17.86% (15/84) | 78.12% (25/32) | 11.11% (2/18) |
| Hyperbilirubinemia | 6% (3/53%) * | 9.68% (3/31) | - | 3.12% (1/32) | - |

The most common side effects seen in the selected studies are shown in the table above. A few other side effects, like thrombocytopenia, were seen in (1) Trastuzumab deruxtecan 10.96% (8/73), (2) Trastuzumab emtansine plus pertuzumab 9.68% (3/31), and (3) Trastuzumab plus pyrotinib 16.67% (3/18). And pruritus was seen in (1) Trastuzumab emtansine plus pertuzumab 9.68% (3/31) and (2) Trastuzumab plus lapatinib 9.37% (3/32). * Trastuzumab deruxtecan had two trials. One trial had 53 patients and reported 28% diarrhea and 6% hyperbilirubinemia. The other trial, with 20 patients, had no information about these side effects. ** Dermatitis was a side effect shown in the second trial of trastuzumab deruxtecan with 20 patients. '-' Either no information or negligible information about this side effect in the trial was given.

## 4. Discussion

Colorectal carcinoma is associated with significant morbidity and mortality, underscoring the need for new therapeutic interventions based on prior treatment knowledge. While HER2-positive overexpression/mutations are only present in 3–5% of cases of metastatic colorectal carcinoma, recent targeted therapies using monoclonal antibodies and ADCs have made HER2 a promising target for research.

This meta-analysis is the first to investigate the efficacy of HER2-targeted therapies for HER2-positive CRC using data from recently completed and ongoing clinical trials. All trials included patients with heavily pre-treated metastatic CRC who received the trial regimen after the confirmation of HER2 receptor positivity via immunohistochemistry (IHC) or fluorescence in situ hybridization (FISH). The MOUNTAINEER-03 trial investigating

tucatinib with trastuzumab and mFOLFOX6 versus mFOLFOX in first-line HER2-positive mCRC is ongoing, and we are awaiting the results [36]. The common element in all the drug combinations discussed here is trastuzumab: A monoclonal antibody against the HER2 receptor that inhibits its downstream effects. Trastuzumab was not given as monotherapy but was conjugated with other chemotherapeutic drugs to enhance anti-cancer effects. The combinations included pyrotinib, tucatinib, and lapatinib, which are TKIs, and trastuzumab deruxtecan, an ADC using deruxtecan, and a topoisomerase agent. Another ADC that was tested was trastuzumab emtansine (DM1 cytotoxic agent), which was investigated in combination with pertuzumab [30].

This analysis of studies identified trastuzumab deruxtecan as the most effective anti-HER2 agent, with the highest DCR (more than 80%) and ORR [29]. Although the percentage of complete responders was relatively low for all drug combinations, trastuzumab plus tucatinib had a comparatively better value of 3.6% [35]. Adverse effects were predictable and commonly seen with most chemotherapeutic agents, with grade 1–2 fatigue, nausea, vomiting, diarrhea, and dermatitis observed commonly in most of those trials. Cytopenia, particularly thrombocytopenia, was common with ADC regimens in trastuzumab deruxtecan and trastuzumab emtansine plus pertuzumab trials. The latter trial had a relatively high DCR with a good safety profile.

The meta-analysis results provide essential insights into the efficacy of HER2-targeted treatment regimens in patients with HER2-positive mCRC. The pooled ORR and DCR of 31.33% and 74.37%, respectively, indicate a significant response to HER2-targeted therapy in this patient population. Additionally, the pooled PFS of 6.2 months suggests that HER2-targeted treatment regimens are associated with a meaningful improvement in survival outcomes. These findings have significant clinical implications and are supported by the recently accelerated FDA approval of the tucatinib plus trastuzumab regimen for RAS wild-type HER2-positive unresectable or mCRC [35]. This highlights the efficacy of HER2-targeted therapies as an effective targeted treatment option for patients with HER2-positive mCRC.

Despite these encouraging findings, it is important to note that statistical heterogeneity was observed in the analysis of ORR and DCR. This heterogeneity may be due to differences in study design, patient characteristics, or other factors not accounted for in this analysis. The main limitation of this estimate is that it always centers the 95% CI around the point estimate and essentially assumes a symmetric distribution. Further research is needed to better understand the factors contributing to the observed variability and type of distribution in response to HER2-targeted therapy in patients with HER2-positive mCRC.

Most studies in our analysis did not examine outcomes in HER2 low-expression variants. The HERACLES diagnostic criteria utilized for colorectal cancer (all tumors expressing 3+ HER2 score in >50% of cells using immunohistochemistry or expressing 2+ HER2 score with a HER2:CEP17 ratio >2 in >50% of cells using FISH) were used to select patients for trial enrollment in three studies including Fu et al., Bianchi et al. and Tosi et al. [30,31,33]. In their study, Bianchi et al. compared PFS and the objective response of patients with HER2-positive tumors based on their HER2 expression levels, suggesting that tumors with a higher HER2 IHC score had better PFS than those with a lower score [30]. The DESTINY CRC-01 trial in HER2-negative patients showed no response among IHC2+/ISH− or IHC1+ mCRC patients. Due to the low number of patients enrolled, these results were not statistically significant, and further studies are necessary to improve the analysis's validity in HER2-low mCRC.

Resistance against HER2-targeted treatments in the initial and acquired stages is common across tumor types [21,37,38]. Within other key effectors of tumorigenesis, molecular alterations, including RAS, PIK3CA, and BRAF mutations, could remunerate for the inhibition of HER2, causing resistance [21,39]. Targeting these alterations sequentially post progression on HER2 targeted regimens or concurrently should be considered in future trials to optimize the overall survival outcomes with those regimens.

Most studies in our analysis excluded RAS mutant patients. Only three trials by Fu et al., Tsurutani et al. and Siena et al. included RAS mutant patients [29,32,34]. Patients with a dual HER2-positive and RAS-mutant status showed minimal changes in their tumor size from the baseline [32]. This subgroup of patients had an ORR of 0%, suggesting that RAS predicts no clinical response to HER2-targeted regimens. Targeting RAS-mutant mCRC remains one of the most difficult challenges in GI oncology, but several KRAS inhibitory agents are being developed and investigated [40]. Future trials may consider combining RAS-targeted approaches with HER2-targeted regimens as a potentially effective strategy in this small subset of patients.

HER2 overexpression/mutation's prognostic utility remains unclear in colorectal cancer. However, mutations or the amplification of HER2 in these cancers are becoming more readily identified due to the increasing use of next-generation sequencing (NGS). We expect that mapping the genomic landscape and the signaling network of HER2-amplified tumors could provide a foundation understanding clonal evolution, tumor heterogeneity, and resistance to HER2-directed therapies.

Regarding our study limitations, the single-arm design of the analyzed studies lacks a control group, and randomization is one limiting feature. Another limitation is the small sample size of these trials, which is too small to allow subgroup analysis to compare therapeutic effects across HER2 levels, HER2 alterations, and the status of other proto-oncogenes. Despite limitations in the selection and drug administration process observed in the analyzed trials, these results highlight the potential of HER2-targeted therapy as an effective treatment option for patients with HER2-positive mCRC and underscore the need for continued research in this area.

## 5. Conclusions

In conclusion, this meta-analysis on HER2-targeted therapies demonstrates the potential for a targeted and effective treatment option for patients with HER2-positive mCRC. The results indicate a significant response to HER2-targeted treatment in this patient population with a meaningful improvement in survival outcomes. These findings are particularly noteworthy given the limited treatment options available for this patient population and the historically poor prognosis associated with this disease. Further research is needed to better understand the factors contributing to observed variability in the response to HER2-targeted therapy and to improve this analysis's validity in patients with HER2-positive mCRC.

**Author Contributions:** Conceptualization, A.C., M.B. and A.S.; methodology, A.C.; validation, A.C., F.A. and M.B.; formal analysis, A.C.; investigation, M.B.; resources, A.S.; data curation, M.M. and R.A.; writing—original draft preparation, A.C. and M.B.; writing—review and editing, F.A., J.Z. and A.S.; visualization, I.H.S. and D.H.; supervision, S.G.; project administration, A.S. All authors have read and agreed to the published version of the manuscript.

**Funding:** This research received no external funding.

**Institutional Review Board Statement:** Not applicable.

**Informed Consent Statement:** Not applicable.

**Data Availability Statement:** Data are available in a publicly accessible repository.

**Conflicts of Interest:** Anwaar Saeed: Consulting or Advisory Role—AstraZeneca; Bristol-Myers Squibb; Daiichi Sankyo/Astra Zeneca; Exelixis; Five Prime Therapeutics; Pfizer. Research Funding—Actuate Therapeutics (Inst); Astellas Pharma (Inst); AstraZeneca/MedImmune (Inst); Bristol-Myers Squibb (Inst); Clovis Oncology (Inst); Daiichi Sankyo/UCB Japan (Inst); Exelixis (Inst); Five Prime Therapeutics (Inst); innovent biologics (Inst); KAHR Medical (Inst); Merck Sharp & Dohme (Inst); Seagen (Inst). Sepideh Gholami: Consulting Role: Iota and Gentech Inc. Ibrahim Halil Sahin: Consulting and Advisory Role: Seagen and GlaxoSmithKline, Lumanity. Akshit Chitkara, Muhammad Bakhtiar, Dennis Hsu, Janie Zhang, FNU Anamika, Mahnoor Mahnoor, and Rabeea Ahmed declare no conflict of interest.

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
