# Peer review of "A Meta-Analysis to Assess the Efficacy of HER2-Targeted Treatment Regimens in HER2-Positive Metastatic Colorectal Cancer (mCRC)"

_curroncol, doi:10.3390/curroncol30090600_

Round 1

Reviewer 1 Report

The study from Akshit Chitkara et al systematically reviewed the efficacy of HER2-targeted treatment regimens in HER2-positive metastatic colorectal cancer. This meta-analysis revealed that a median of three prior lines of therapy was used before enrolling in the six trials comprising 238 patients with HER2-positive metastatic colorectal cancer. The pooled objective response rate and disease control rate 28 were 31.33% (95% confidence interval [CI] 24.27-38.39) and 74.37% (95% CI 64.57-84.17), respectively. The pooled weighted progression-free survival was 6.2 months. The pooled ORR and DCR meta-analysis indicate a significant response to HER2-targeted therapy in HER2-positive mCRC patient with improvement of their survival outcomes. This study increases our understanding for the efficacy of HER2-targeted treatment in HER2-positive metastatic colorectal cancer

Here, we have several concerns need to be addressed.

1. The enrolled studies did not have a reference in Table 2.

2. As a meta-analysis, the introduction and discussion part included too less reference. Usually, it should be at least 50-60 reference for this kind of paper.

NA

Author Response

Dear Reviewer,

Thank you for your thorough review and positive feedback on our manuscript. We are pleased to hear that the content aligns with the scope of the journal and is of interest to its broad readership. We appreciate the constructive comments and have addressed each one as detailed below:

Moderate editing of English language required

  • Language has been professionally edited and revised for correct grammar and better flow. Please review the revised manuscript.

Does the introduction provide sufficient background and include all relevant references? Must be improved

  • The introduction has been revised wherever appropriate, and multiple references have been added to support the text.

Are all the cited references relevant to the research? Must be improved

  • All citations have been thoroughly revised and checked for appropriateness.

Is the research design appropriate?Can be improved

  • The research design and methods were finalized after a thorough discussion and in-depth study of the meta-analysis of single-arm trials done in the past. We have studies using similar designs and methodologies which have been published. We have also worked on simplifying and cleaning the format flow in this study.

Are the results clearly presented? Can be improved

Are the conclusions supported by the results? Can be improved

  • We have cleaned up the results section and simplified the table formatting for better understanding to support our conclusion. Also, added appropriate references for the studies.

  1. The enrolled studies did not have a reference in Table 2.

 References have been labeled in Table 2.

  1. As a meta-analysis, the introduction and discussion part included too less reference. Usually, it should be at least 50-60 reference for this kind of paper.

Being a relatively newer subcategory of treatment options for mCRC, there aren't many references that fit into our Introduction and Discussion. Adding less relevant references will dilute the relevance of the existing literature references selected for our study. Nonetheless, we have added more references wherever appropriate for our meta-analysis.

We would like to express our sincere gratitude for your insightful comments and suggestions. We believe that the revisions have further enhanced the quality of our manuscript, and we hope that the revised version meets with your approval.

Sincerely,

Akshit Chitkara on behalf of all authors.

Reviewer 2 Report

The meta-analysis was conducted following the Cochrane Handbook and written following the PRISMA guidelines. After a comprehensive search for relevant articles, 14 CTs were identified and uploaded to Rayyan, and six trials were ultimately selected for inclusion.

The meta-analysis presented for review is an attempt to determine the efficacy of treating HER2-positive metastatic colorectal cancer (mCRC) with HER2-targeted therapy. The paper ultimately analyses six papers by other authors, but which are not (except for one - Fu et al.) cited in the references. The results of these papers are cited and commented on in the Results subsection, but the reader cannot look at them, e.g. Bianchi et al. (line 196), Siena et al. (line 201), Tsurutami et al. (line 208), etc. Statistical data are included only in Table 4.

Already in Table 1, there should be citation numbers next to the names and these papers included in the literature list. In the other Tables, only the citation numbers can remain.

In the Discussion subsection it is stated that the most effective anti-HER2 agent is trastuzumab deruxtecan, with the highest ORR and DCR above 80%. From Table 4 it appears that only the DCR is above 80%, as the ORR varies between 15-46%. Please verify this.

Other comments:

1. there is some confusion in the citations. In the Introduction of the paper, items 8 and 9 (lines 52 and 54) are cited before items 6 and 7 (lines 56-58). This needs to be corrected.

2. There is a complete absence of citations of papers 13-16 in the text, and immediately there is paper 17 (line 144). Item 16 i.e. Fu et al. is included in the literature but not in the text. Please correct this, as it is not understandable.

3. Table 3 is not cited in the text.

4. Every table needs a legend, with an explanation of even the simplest abbreviations (e.g. N/A, CR, PR, SD, PD, as it needs to be self-sufficient.

5. there are some editorial errors in the paper (no full stop at the end of the sentence in line 84), a break in Table 1, thrombocytopenia with a lowercase letter in line 246; unnecessarily repeated explanations of abbreviations (e.g. ORR, DCR in lines 109-110 and 113-114, and also in 135-136; add the word "mutations" (?) after citations of RAS, PIK3CA, and BRAS (line 320). Under Table 5, first give explanations of the Table in the form of *, **, "-" and then "A few other..."

Once the suggested corrections have been completed, the paper can be accepted for publication.

Author Response

Dear Reviewer,

Thank you for your thorough review and positive feedback on our manuscript. We are pleased to hear that the content aligns with the scope of the journal and is of interest to its broad readership. We appreciate the constructive comments and have addressed each one as detailed below:

 English language fine. No issues detected

Thank you for appreciating and approving the English language and flow.

Does the introduction provide sufficient background and include all relevant references? Can be improved

Are all the cited references relevant to the research? Must be improved

  • We worked on adding more references to the introduction. We have cleaned up the references based on our research.

Is the research design appropriate?Can be improved

  • The research design and methods were finalized after a thorough discussion and in-depth study of the meta-analysis of single-arm trials done in the past. We have studies using similar designs and methodologies, which have been published. We have also worked on simplifying and cleaning the format flow in this study.

Are the results clearly presented? Must be improved

Are the conclusions supported by the results? Can be improved

  • We have cleaned up the results section and simplified the table formatting for better understanding to support our conclusion. Also, added appropriate references for the studies.

The paper ultimately analyses six papers by other authors, but which are not (except for one - Fu et al.) cited in the references.

  • We have added all the included trials in the references and cited wherever appropriate.

The results of these papers are cited and commented on in the Results subsection, but the reader cannot look at them, e.g. Bianchi et al. (line 196), Siena et al. (line 201), Tsurutami et al. (line 208), etc. Statistical data are included only in Table 4.

  • The results of the individual trial are added in the table format as statistical data and some of it as part of the text to keep the article reader-friendly, concise, and to the point of relevance. Adding all detailed results from individual studies to the text will cause data cluttering and confusion. Our study aims to generate pooled data and simplify the results.

Already in Table 1, there should be citation numbers next to the names and these papers included in the literature list. In the other Tables, only the citation numbers can remain.

  • We have added the references to study titles in Tables 2-4. We will keep the study names for easy referencing by the reader when going through the results and discussion.

In the Discussion subsection it is stated that the most effective anti-HER2 agent is trastuzumab deruxtecan, with the highest ORR and DCR above 80%. From Table 4 it appears that only the DCR is above 80%, as the ORR varies between 15-46%. Please verify this.

  • Restructured the sentence to clarify the highest DCR of more than 80% and the highest ORR.

Other comments:

  1. there is some confusion in the citations. In the Introduction of the paper, items 8 and 9 (lines 52 and 54) are cited before items 6 and 7 (lines 56-58). This needs to be corrected. References have all been arranged in sequence as suggested.

  1. There is a complete absence of citations of papers 13-16 in the text, and immediately there is paper 17 (line 144). Item 16 i.e. Fu et al. is included in the literature but not in the text. Please correct this, as it is not understandable. All references have been cited in the text.

  1. Table 3 is not cited in the text. Table 3 has been cited in the text.

  1. Every table needs a legend, with an explanation of even the simplest abbreviations (e.g. N/A, CR, PR, SD, PD, as it needs to be self-sufficient. Legends have been added and elaborated wherever appropriate.

  1. there are some editorial errors in the paper (no full stop at the end of the sentence in line 84), a break in Table 1, thrombocytopenia with a lowercase letter in line 246; unnecessarily repeated explanations of abbreviations (e.g. ORR, DCR in lines 109-110 and 113-114, and also in 135-136; add the word "mutations" (?) after citations of RAS, PIK3CA, and BRAS (line 320). Under Table 5, first give explanations of the Table in the form of *, **, "-" and then "A few other..." Editorial errors have been addressed."

We would like to express our sincere gratitude for your insightful comments and suggestions. We believe that the revisions have further enhanced the quality of our manuscript, and we hope that the revised version meets with your approval.

Sincerely,

Akshit Chitkara on behalf of all authors.